# An Observational Case-Control Study on Parental Age and Childhood Renal Tumors

**DOI:** 10.3390/cancers15215144

**Published:** 2023-10-26

**Authors:** Georgios Politis, Stefan Wagenpfeil, Nils Welter, Marvin Mergen, Rhoikos Furtwängler, Norbert Graf

**Affiliations:** 1Department of Pediatric Oncology and Hematology, Saarland University, 66421 Homburg, Germany; georgios.politis@uks.eu (G.P.); nils.welter@uks.eu (N.W.); marvin.mergen@uks.eu (M.M.); rhoikos.furtwaengler@uks.eu (R.F.); 2Institute of Medical Biometry, Epidemiology and Medical Informatics (IMBEI), Saarland University, 66421 Homburg, Germany; stefan.wagenpfeil@uks.eu

**Keywords:** Wilms tumor, parental age, childhood renal cancer, incidence and outcome

## Abstract

**Simple Summary:**

Wilms tumor (WT), as the most common renal tumor in childhood, is treated very successfully within prospective trials and studies. The association with various genetic alterations has been studied, but it is still unclear why a WT develops in a specific child and why some children suffer from relapse. Genetic risk and external, environmental factors need further attention. Parental age at birth plays a role in various childhood diseases, so it is reasonable to also investigate whether older parental age is a risk factor for the development of childhood renal tumors. We could demonstrate that parental age has no correlation with the incidence of renal tumors in childhood.

**Abstract:**

Despite excellent outcomes, many open questions remain about Wilms tumor (WT). Influences and risk factors for tumorigenesis, as well as tumor aggressiveness and recurrence, are not fully understood. Parental age plays a role in various childhood diseases and is also discussed as a risk factor for childhood cancer. We analyzed both maternal and paternal age at birth as risk factors for the occurrence of Wilms and non-Wilms tumors in children and investigated whether older maternal or paternal age is associated with a higher tumor incidence. During 1990 and 2019 we collected data from 3991 patients from the multicenter studies SIOP9/GPO, SIOP 93-01/GPOH, and SIOP 2001/GPOH, of whom maternal and paternal age was available in 2277 cases. Data from the Federal Statistical Office containing live births in Germany from 1990–2019 served as a comparative database. For maternal age at birth, the control data yielded 22,451,412 cases and for paternal age yielded 19,046,314 cases. Comparing maternal and paternal ages of the study patients with those of the control data, we confirmed that higher parental age is not correlated with the incidence of renal tumors in childhood. Mean ages of fathers and mothers in patients and the control cohort increased between 1991 and 2019 (fathers: 30.28 vs. 34.04; mothers: 27.68 vs. 29.79 in the patient group and 31.29 vs. 34.23 and 28.88 vs. 32.67 in the control group, respectively) without higher numbers of patients with kidney cancer over time. No influence was found for the subtype of cancer nor for syndromes. In addition, overall survival of patients is independent of the year of diagnosis and the age of the parents but depends on histology type and stage in WT.

## 1. Introduction

As the most common renal tumor in childhood, Wilms tumor (WT) has been successfully treated for more than 50 years due to prospective multicenter trials and studies [1]. Research on the etiology of nephroblastoma is still ongoing. Many genes associated with nephroblastoma have been identified [2], but the molecular genetic background is not known for all patients. In addition, it is unclear which external factors, if any, promote the development of childhood renal tumors.

Parental age plays a role in several other childhood diseases and is also repeatedly discussed as a risk factor for childhood cancer. In this work, the age of both mothers and fathers at the birth of their children with WT is analyzed as a risk factor for the occurrence of WT and non-WT [3,4]. This may have an influence on tumor screening in children of parents beyond a possible age threshold with a higher risk for WT.

Several research groups have already tried to answer this question for WT. Results among the studies are different, so findings regarding the risk of parental age on the incidence of WT remain inconclusive (Table 1). External, environmental factors related to the region where the child is born may overlap with the risk of the age of parents, as older parents are longer exposed to environmental factors. Corresponding results are reviewed in our work on a larger patient cohort in comparison with the general German population. In this work, we neglected the influence of different regions in Germany and concentrated only on the age of parents to analyze the impact of parental age on the occurrence of WT in children, on the different histology of renal tumors, and on outcome.

## 2. Materials and Methods

This observational case-control study with unmatched controls comes from several prospective, multicenter trials and controls from a dataset derived from the entire population of live births in Germany describing this population. During the trials SIOP9/GPO, SIOP 93-01/GPOH, and SIOP 2001/GPOH, the ages of parents at the time of the birth of their child with a kidney tumor were stored in the respective databases. During 1989–2020, these trials enrolled 3991 patients from Germany, Austria, and Switzerland in whom we could retrieve data on age for 2277 (57.05%) mothers and fathers since 1990 from Germany (data set of cases). Besides parental age, patient data (age, gender, syndromes, tumor histology, local and overall stage, surgery, chemotherapy, radiotherapy, complications, and outcome) were taken into consideration. Only German kidney tumor patients after live birth with centrally reviewed histology were included, and characteristics of these patients are given in Appendix A. Data of the control group were provided anonymously by the Federal Statistical Office of Germany [11]. These data included the age of mothers and the age of fathers, of all live births in Germany. Further processing of the control group’s data was needed and carried out with the help of Microsoft^®^ Excel^®^ Version 16.63.1 (22071301); 64bit for Mac. To compare the patient data (cases) with the control data (controls), we only included patients that were born between 1990 and 2019 in Germany. Data about the region where the children with kidney tumors were born were not included in this analysis. For the control group, we considered the years 1990–2019 for maternal age and the years 1991–2019 for paternal age. In this group, exact case numbers below 16 years and above 49 years for mothers were missing and were excluded as well as those with unknown maternal age. Analogous to mothers, we also excluded data for fathers where the age was unknown or no exact numbers below 15 or above 54 years of age were given. Those excluded data were less than 600/year (max.: 0.1%) in women and less than 2200/year (max.: 0.3%/year) in men. Altogether, 22,451,412 maternal and 19,046,314 paternal cases were included in the control data set. It is unknown how many children with kidney cancer will be born after 2019 in the control group. Considering the number of live births/year in Germany as given by the Federal Statistical Office of Germany [11] (max: 809,019 in 1991 and min.: 662,685 in 2011 during the analyzed time-period) and the number of children born with kidney cancer/year in Germany (around 100/year), this number is less than 0.02%/year. These restrictions given by the data sets are neglectable for statistical analysis. 

Comparing the number of patients with WT enrolled in this analysis with the number of patients with WT from the German Childhood Cancer Registry, 99.4% were enrolled in the clinical trials between 2008 and 2018 (https://www.kinderkrebsregister.de/, accessed on 2 October 2023), making our study cohort representative for patients with WT [12].

All data were pseudonymized before statistical analysis and treated according to the general data protection regulation (GDPR) of the EU. Computational and statistical analysis was performed using SPSS 27 for Mac (IBM SPSS Statistics 25.0.0.2; 64bit; Armonk, NY, USA). Qualitative and quantitative values are presented as relative and absolute frequencies as well as mean ± standard deviation. T-test for 2 independent samples and Welch’s *t*-test were used to compare means between two independent groups. Survival was evaluated by Kaplan-Meier analyses as well as Cox regression analysis. Two-sided significance was defined as *p* < 0.05 for all the statistical tests. Furthermore, in the literature overview odds ratios and respective 95% confidence intervals are given. We did not account for the issue of multiple statistical tests due to the explorative nature of the investigation and the very large sample size in the controls. 

All clinical trials were reviewed and approved by the Ethics Committee of the Medical Association of Saarland (/LS of 23/04/1993, no. 136/01 of 20/09/2002 and 248/13 of 13/01/2014). 

## 3. Results

Mothers of patients included in the study (cases) were between 15 and 46 years of age (mean age: 29.68 years, standard deviation (STD): 5.393 years), and fathers were between 17 and 69 years of age (mean age: 32.77 years, standard deviation: 6.39 years). The control data set consisted of a total of 22,451,412 females aged 16–49 years and 19,046,314 males aged 15–54 years. The age distributions and descriptive statistical parameters for cases and controls are shown in Figure 1.

The comparison between cases and the controls did not show a significant difference in the mean age of mothers (29.68 ± 5.39 (cases) vs. 29.83 ± 5.36 (controls); *p* > 0.05). This was different for paternal age, where fathers of children with renal tumors were significantly younger than those of the control data set (mean age 32.77 ± 6.39 vs. 33.07 ± 5.95 years of age; *p* < 0.05). Despite the significant difference for fathers, it can be concluded that mothers and fathers of children with renal tumors are not older than the parents of the general German population (control data set) at the time of the birth of their child (Table 2).

### 3.1. Correlation between Maternal and Paternal Age in Patients with WT

The correlation between maternal and paternal age was only possible to analyze in the patient group. This information is lacking in the controls as there was no link between the age of the mother and the father of a single child. Fathers are, in mean, 2.96 ± 4.76 years older (min = −18 and max = 32 years) than mothers of a child with WT. The distribution of age differences between mothers and fathers for children with WTs is shown in Figure 2.

### 3.2. Maternal and Paternal Age over Time

Maternal and paternal age increased during the analyzed time period in the control group and the patient group in fathers and mothers. In the patient group, the mean parental age was significantly lower in 1991 (30.28 years) compared to 2019 (34.04), and in the control group, it was 31.29 years in 1991 compared to 34.23 years in 2019. For mothers, the mean ages were 27.68 in 1991 and 29.79 years in 2019 and 28.88 and 32.47 years, respectively. No differences were found between the mean ages of cases and controls during the whole study period for either fathers or mothers. Figure 3 shows the mean adjusted age of two 15-year time periods (1990/1991–2004 and 2005–2019) for fathers and mothers in both cohorts.

### 3.3. Different Histology

For the whole group of WTs, there is no difference between the mean ages of fathers and mothers compared between cases and controls. Despite the collection of data over a 30-year period, the number of WTs per histological subtype is low and makes statistical analysis difficult (Table 3). Stromal (*n* = 202), mixed (*n* = 560), and regressive types (*n* = 568) were those with the largest number with known maternal age at birth. Only in the mixed type were mothers of children with WT significantly younger than in the control group (29.34 ± 5.55 vs. 29.83 ± 5.36 years; *p* = 0.03). All other histological WT types did not show a significant difference in the mean age between both cohorts. For WT patients with known paternal age, most cases were also seen in stromal (*n* = 201), mixed (*n* = 500), and regressive types (*n* = 554). There was no significant difference in any of the different histological subgroups for WT. Thus, mothers and fathers of children with any WT are not older than the parents of the general German population (Table 3). In the case of non-WTs, the number of patients is too low to draw any conclusions.

### 3.4. Different Syndromes

Because nephroblastoma in children often occurs in association with specific syndromes, we specifically compared cases with controls. In none of the most common syndromes a significant difference in maternal or paternal age at the birth of their child with WT was found. This is true for the cohort of all patients with a syndrome as well as for the different syndromes analyzed (Table 4). However, it needs to be considered that the number of WT patients with a specific syndrome is small, and only large differences in parental age between the patient and the control group can be detected.

### 3.5. Outcome

To answer the question if the age of parents is influencing the outcome of children with renal tumors, we analyzed separately the age of mothers and fathers of deceased children against controls. For mothers, the group of deceased children showed a mean age of 29.10 years, and for fathers, it showed a mean age of 31.97 years at birth of their child (Table 5). In none of them was a significant difference found, underlining that the age of parents has no impact on the outcome of children with renal tumors. In WT only, this was also confirmed by Cox regression for overall survival considering the year of diagnosis, the age of fathers and mothers, and the year of birth of the child as parameters together with the known risk factors of the child’s age at diagnosis, histology, and local stage. In this model, only histology and local stage were significant with a Hazard ratio of 3.48 for local stage II and 13.10 for local stage III compared to local stage I and a Hazard ratio of 1.31 for intermediate risk and 4.91 for high risk compared to low risk (Appendix A). In addition, Kaplan Meier analysis for the time of diagnosis and for the age of fathers and mothers showed no significant differences in the log rank test (Appendix A). 

## 4. Discussion

Our results confirm that the age of parents at the birth of children with renal tumors is not different from the age of parents at the birth of their child in the general German population for fathers or mothers. As case numbers were below 100 for every non-Wilms tumor type, only large differences in parental ages compared with the general population could likely be detected. Therefore, this discussion is restricted to Wilms tumor only. In addition, fathers and mothers were not older than the general population for all types of renal tumors in childhood and in patients with syndromes. No such data were available in the literature up to now. Furthermore, the outcome of children with renal tumors is not influenced by the age of the parents or the age difference between fathers and mothers at the time of the birth of their child with a renal tumor.

To the best of our knowledge, our cohort of parents of children with renal tumors is the largest published so far (Table 1). Those with larger numbers of cases [7,8,10] included not only renal tumors but also childhood cancer in general. Yip et al. [10] confirms our findings of no association between the age of parents and the occurrence of Wilms tumors in children as is also the case in a study with fewer patients [6]. In a Danish population-based registry [8], it is pointed out that further analyses of biological and social factors need to be done to explain a slightly increased risk of Wilms tumor with older paternal age. This was also seen by Olson et al. [5], who found a positive relationship between paternal age and Wilms tumor incidence. Such increased risk may support epigenetic factors for the development of Wilms tumors. In contrast, a pooled analysis by Johnson et al. [7] observed a positive linear trend per 5-year maternal age increase for WT but not for paternal age. Also, Sharpe et al. [9] detected such a risk for WT with increasing maternal age but only in children diagnosed before 25 months of age. 

In summary, parental age as a risk factor for WT cannot be assessed clearly. One reason for the different results is linked to the composition of the patient cohorts and the control groups. It is important that our cases represent a nationwide population over 30 years, which is the largest published series so far.

In this context, Sharpe et al. [9] stated that it may be useful to investigate more details to exclude environmental factors after birth as possible relevant risk factors. This also includes the intrauterine influence, which is strongly dependent on the mother and her behavior. The time and intensity of the influences are important parameters. After years of exposure, environmental factors will have considerably more indirect influence on children in older than in younger parents through epigenetic changes. A study in Brazil by Sharpe et al. [13] showed that occupational exposure to pesticides may favor the occurrence of Wilms tumors. Agricultural work with frequent pesticide used by both mothers and fathers has been shown to increase the risk of disease in their children. It is well known that parents’ exposure to external factors such as stress, education, social status, place of residence, noxious substances, diet, and environmental influences also has a consecutive effect on unborn children. This has been discussed in several studies considering such influences of environmental exposures [14] like pesticides [15], hydrocarbon solvents and engine exhausted fumes [16], intake of vitamins and folic acid as well as drinking or smoking during pregnancy [17], maternal use of household pesticides and insecticides [18], hazardous chemicals [19], paternal lead (Pb) exposure [20], and paternal occupation exposures [21].

As the incidence of WT in Germany did not change over time according to the German Cancer Registry [12], it is of interest to see that the age of parents in our control group increased over time. As given in Figure 2, the mean age of mothers increases significantly from 28.80 to 30.31years and that of fathers increases from 31.70 to 33.41 years comparing the two time periods 1990–2004 and 2005–2019. This rise in the age of the parents of a child with WT is not different than for fathers and mothers in the control group. At least in Germany, a longer exposure to environmental factors did not increase the risk for childhood renal tumors. As we have no data of residence for our controls, we cannot exclude differences in the number of patients with WT in smaller areas in Germany depending on the age of their parents. Therefore, further studies are needed that consider and analyze environmental factors and the residency of patients and the family in addition to parental age.

## 5. Conclusions

According to our observational case cohort study, we could not detect an increased risk of the development of a WT in a child attributable to parents’ older age. However, there are inconclusive answers to this question in the literature. Environmental factors may play a role, especially after longer exposure time. Therefore, further studies are needed that consider and analyze environmental factors in addition to parental age.

## Figures and Tables

**Figure 1 cancers-15-05144-f001:**
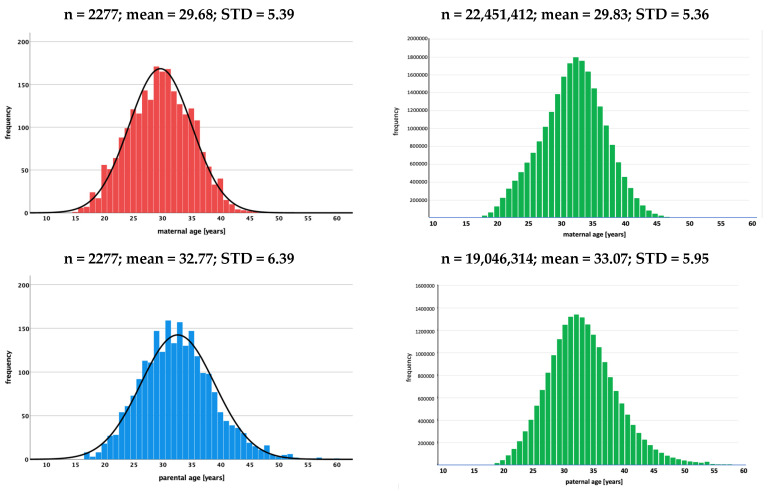
Histograms showing the age distribution of mothers and fathers at birth of children with renal tumors (study data) and of females (1990–2019) and males (1991–2019) at the time of birth of their child in the control group (live births in Germany) [11]. (n: number of cases, mean: mean age, STD: standard deviation).

**Figure 2 cancers-15-05144-f002:**
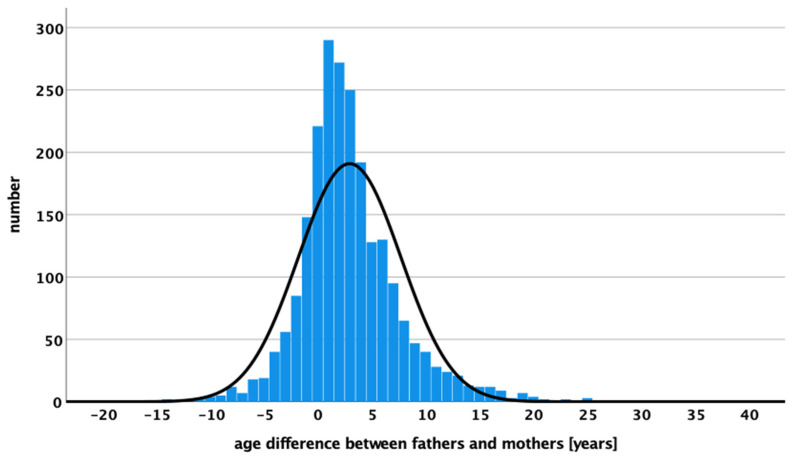
Age differences in years between mothers and fathers of children with WT are displaying the number of WT cases.

**Figure 3 cancers-15-05144-f003:**
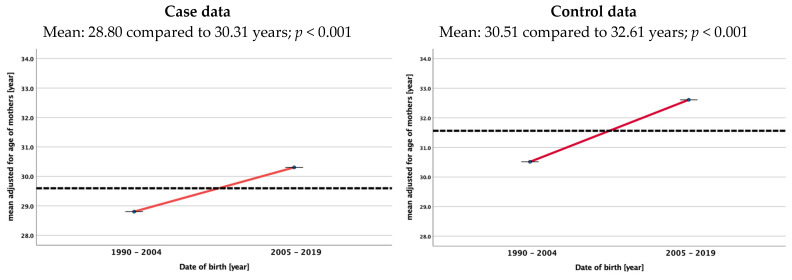
Mean adjusted age of mothers and fathers at birth of children with renal tumors (study data) and of females and males at the time of birth of their child in the control group (live births in Germany) [11]. The dotted lines always represent the mean age over the whole study period.

**Table 1 cancers-15-05144-t001:** Literature overview.

Authors	Study Title	Cases/Years	Results
Olson J, et al. (1993) [5]	Wilms’ tumour and parental age: a report from the National Wilms’ Tumour Study	n (2437)1996–1987	Positive relationship between parental age and Wilms tumor incidence, so at least some Wilms tumors may be due to novel germline mutations.
Heuchl J, et al. (1996) [6]	Birth characteristics and risk of Wilms’ tumour: a nationwide prospective study in Norway	n (119)1967–1992	Neither maternal age nor paternal age is associated with increased risk of Wilms tumor.
Johnson, Kimberly J, et al. (2004) [7]	Parental age and risk of childhood cancer: A pooled analysis	n (17,672)1980–2004	Positive linear trend with 5-year increase in maternal age. Observed for children diagnosed with cancer in general (odds ratio = 1.08 [95% confidence interval = 1.06–1.10]) for Wilms tumor (1.16 [1.09–1.22]).
Contreras, Zuelma A., et al. (2017) [8]	Parental age and childhood cancer risk: A Danish population-based registry study	n (5856)1968–2015	The risk of Wilms tumor also appeared to be increased with older paternal age [OR = 1.11, 95% CI: (0.97; 1.28) per 5-year age increment]. Older parental age was a risk factor for several childhood cancers in Danish children. Further investigation of the biological and social factors that may contribute to these associations is warranted.
Sharpe Franco, et al. (1999) [9]	The influence of parental age on the risk of Wilms’ tumour	n (109)1987–1989	For cases diagnosed before 25 months of age, there was a clear gradient of increasing risk of WT with increasing maternal age at the time of the child’s birth. There was no increased risk for cases diagnosed after 25 months of age. The effects of paternal age were less pronounced.
Yip B, et al. (2006) [10]	Parental age and risk of childhood cancers: A population-based cohort study from Sweden	n (7844)1961–2000	Maternal age was not associated with increased risk of Wilms tumor and NHL. No significant association was found between paternal age and NHL or Wilms tumor.

**Table 2 cancers-15-05144-t002:** Maternal and paternal age of the test and control data sets.

		Cases [n]	Mean Age [Years]	STD [Years]
Test data set	Mothers	2277	29.68 *	5.393
	Fathers	2277	32.77 °	6.388
Control data set	Mothers	22,451,412	29.83 *	5.363
	Fathers	19,046,314	33.07 °	5.948

Welch’s *t*-test: *: *p* = 0.18; °: *p* = 0.02; STD: standard deviation.

**Table 3 cancers-15-05144-t003:** Age comparison between cases and controls stratified by gender for different types of WTs and non-WTs.

	Mothers				Fathers			
	n	Mean Age [y]	STD [y]	*p*-Value	n	Mean Age [y]	STD [y]	*p*-Value
**cases**								
*CMN*	*80*	*29.14*	*5.26*	*>0.05*	*80*	*32.59*	*6.57*	*>0.05*
*CPDN*	*25*	*31.64*	*5.50*	*>0.05*	*22*	*32.73*	*6.38*	*>0.05*
**Only WT**	**1846**	**29.76**	**5.37**	**>0.05**	**1756**	**32.86**	**6.30**	**>0.05**
CN	78	29.86	5.29	>0.05	75	32.58	5.69	>0.05
epithelial	120	29.15	5.52	>0.05	119	32.29	6.46	>0.05
stromal	202	30.24	5.36	>0.05	201	33.30	6.65	>0.05
Mixed	560	29.34	5.55	**0.037**	500	32.56	6.40	>0.05
regressive	568	30.04	5.17	>0.05	554	33.05	6.22	>0.05
FA	36	28.75	5.26	>0.05	34	32.74	6.36	>0.05
DA	91	30.73	5.22	>0.05	87	33.63	6.40	>0.05
NR	61	29.93	5.40	>0.05	61	33.56	6.30	>0.05
Blastemal *	130	29.63	5.52	>0.05	125	32.35	5.77	>0.05
*CCSK*	*69*	*29.33*	*5.51*	*>0.05*	*64*	*32.47*	*7.65*	*>0.05*
*MRTK*	*43*	*28.60*	*6.02*	*>0.05*	*42*	*31.88*	*6.64*	*>0.05*
*RCC*	*25*	*28.40*	*5.55*	*>0.05*	*25*	*32.48*	*7.96*	*>0.05*
*Cystic nephroma*	*11*	*28.36*	*7.24*	*>0.05*	*11*	*34.45*	*6.01*	*>0.05*
*Adenoma*	*5*	*31.60*	*2.88*	*>0.05*	*5*	*34.40*	*4.39*	*>0.05*
*Other benign*	*11*	*31.55*	*6.55*	*>0.05*	*11*	*33.64*	*7.15*	*>0.05*
*Other malignant*	*17*	*30.12*	*5.63*	*>0.05*	*16*	*33.38*	*6.01*	*>0.05*
**controls**								
	22451412	29.83	5.36		19046314	33.07	5.95	

STD: standard deviation; y: years; CMN: Congenital mesoblastic nephroma; CPDN: Cystic partially differentiated nephroblastoma; CN: Completely necrotic, FA: Focal anaplasia; DA: Diffuse anaplasia; NR: nephroblastomatosis; CCSK: Clear cell sarcoma of kidney; MRTK: Malignant rhabdoid tumor of the kidney; RCC: Renal cell carcinoma; *: after preoperative chemotherapy. Non-WTs are written in italics. Significant p-values are written in bold as well as the row representing only WTs.

**Table 4 cancers-15-05144-t004:** Age comparison between cases and controls stratified by gender for the most important different syndromes and malformations in childhood renal tumors.

	Mothers				Fathers			
	*n*	Mean Age [y]	STD [y]	*p*-Value	*n*	Mean Age [y]	STD [y]	*p*-Value
**Cases**								
All *	343	29.70	5.49	>0.05	328	32.89	6.22	>0.05
Aniridia	11	31.00	7.01	>0.05	10	35.8	9.57	>0.05
WAGR	15	29.80	6.72	>0.05	15	33.87	8.50	>0.05
GU	60	29.75	4.87	>0.05	57	32.14	6.16	>0.05
DDS	20	29.80	5.55	>0.05	20	34.15	7.03	>0.05
BWS	30	29.67	4.22	>0.05	29	32.14	4.68	>0.05
HH	32	31.06	5.40	>0.05	28	32.79	5.20	>0.05
Familial WT	21	29.67	5.86	>0.05	21	31.76	5.82	>0.05
**Controls**								
	22,451,412	29.83	5.36		19,046,314	33.07	5.95	

STD: standard deviation; y: years; *: patients with syndromes or malformations; WAGR: Wilms tumor-aniridia-genitourinary anomalies–range of developmental delays; GU: genitourinary anomalies; DDS: Denys-Drash syndrome; BWS: Beckwith Wiedemann syndrome; HH: hemihypertrophy.

**Table 5 cancers-15-05144-t005:** Comparison of age of parents of deceased children with renal tumors with the age of parents of the control group.

	Mothers				Fathers			
	N	Mean Age [y]	STD [y]	*p*-Value	*n*	Mean Age [y]	STD [y]	*p*-Value
cases	63	29.10	5.13	>0.05	58	31.97	5.64	>0.05
controls	22,451,412	29.83	5.36		19,046,314	33.07	5.95	

T: test set; C: control set; *p*-value refers to Welsh’s *t*-test.

## Data Availability

The data presented in this study are available on request from the corresponding author. The data are not publicly available due to ongoing analysis.

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
