# Peer review of "An Observational Case-Control Study on Parental Age and Childhood Renal Tumors"

_cancers, 2023, doi:10.3390/cancers15215144_

Round 1

Reviewer 1 Report (Previous Reviewer 1)

Review comments on previous version adequately addressed.

Author Response

Thanks to the reviewer for providing  feedback again. We are happy that we could address previous comments adequately.

Reviewer 2 Report (Previous Reviewer 2)

1. Page 2 Line 50,  an "o" is missing before "f parental age".

2. Figure 1, the mean for the case paternal age is different from that in the text and Table 2.  Also in Figure 1, the ns for two control related figures were not correct.

3. Figure 1 and Table 2 present similar information, just keep one.

4. Figure 3, either draw the study cohort and the control in the same figure or use the same scale for y-axis.  Current figures are a little misleading.

Author Response

Thanks to the reviwer for providing his important comments to the paper. Here are the answers to the reviewer:

  1. Page 2 Line 50,  an "o" is missing before "f parental age": The o is added.
  2. Figure 1, the mean for the case paternal age is different from that in the text and Table 2.  Also in Figure 1, the ns for two control related figures were not correct: Thanks for pointing this out. The numbers are corrected now.
  3. Figure 1 and Table 2 present similar information, just keep one: In our view  figure 1 adds the distribution of the ages very nicely. Table 2 adds the significance to the numbers, why we would like to include figure 1 and table 2.
  4. Figure 3, either draw the study cohort and the control in the same figure or use the same scale for y-axis.  Current figures are a little misleading: This is done.

Reviewer 3 Report (Previous Reviewer 3)

None.

Author Response

Thanks to the reviewer for providing  feedback again. We are happy that we could address previous comments adequately.

Reviewer 4 Report (Previous Reviewer 4)

I appreciate the changes made in response to the previous review.

I have just a few small suggested changes mostly in the introduction.

In line 48 you refer to the parents of advanced age as 'elderly'. My understanding is that this is no longer the preferred term, but rather 'older adults'. However, in this context, it may be better to refer to a specific age cutoff. The CDC refers to individuals ages 65 and up as older adults. And with the upper age of your participants is only in the mid-50s, I am not sure that this term is appropriate. I would refer to an age cutoff for what you mean by 'elderly' and drop the term altogether.

Line 50 says 'f' but I think it should be 'of'.

I find the sentence spanning lines 51-52 to be confusing. Do you mean environmental factors may interact with parental age?

Line 61-2 remove 'came from'.

Line 68 - still refers to ‘test data set’.

Line 99 - does the SPSS citation match the version used?

Author Response

Thanks to the reviwer for his valuable comments. Here are our anywres:

I appreciate the changes made in response to the previous review. --> We thank the reviewer for this staement.

I have just a few small suggested changes mostly in the introduction.

In line 48 you refer to the parents of advanced age as 'elderly'. My understanding is that this is no longer the preferred term, but rather 'older adults'. However, in this context, it may be better to refer to a specific age cutoff. The CDC refers to individuals ages 65 and up as older adults. And with the upper age of your participants is only in the mid-50s, I am not sure that this term is appropriate. I would refer to an age cutoff for what you mean by 'elderly' and drop the term altogether: --> Thanks for this comment. It is difficult to define a specific age cut-off without knowing if such a threshold of age exists. Therefore, we have changed the sentence as follow: "This may have an influence on tumor screening in children of parents beyond a possible age threshold with a higher risk for WT."

Line 50 says 'f' but I think it should be 'of'.: This is changed. Thanks for popinting this out.

I find the sentence spanning lines 51-52 to be confusing. Do you mean environmental factors may interact with parental age? --> No, we want to say that the older parents are the longer they were exposed to environmental factors. We have changed the sentence to: "External, environmental factors related to the region, where the child is born, may overlap the risk of the age of parents, as older parents are longer exposed to environmental factors.."

Line 61-2 remove 'came from'. --> Thanks fro pointing this out. The two words are deleted.

Line 68 - still refers to ‘test data set’. --> Thanks for pointing this out. This is changed to "data set of cases".

Line 99 - does the SPSS citation match the version used?

This manuscript is a resubmission of an earlier submission. The following is a list of the peer review reports and author responses from that submission.

Round 1

Reviewer 1 Report

It is still an open question whether risk of a child developing a renal tumour, especially Wilms tumour, varies with the age of either or both parents.  The manuscript under review aims to investigate this by comparing parental ages at the time of child’s birth among a large series of children included in clinical trials for childhood renal tumours in Germany with parental ages in the population of children in Germany as a whole.  The study series consisted of children diagnosed during 1990-2020 from three successive clinical trials.    A strength of the study is the large number of children with Wilms tumour included.  However, the study also suffers from several serious flaws.  These and other points are considered in the specific comments below.

1. The German patients with information on maternal age and paternal age represented 57.05% and 54.52% respectively of all children enrolled from Germany, Austria and Switzerland during 1989-2020.  These percentages are irrelevant to assessing the proportion of eligible children for whom the information on parental age was available.  The denominators should be the numbers enrolled from Germany only starting in 1990 (for maternal age) and in 1991 (for paternal age) respectively.

2.  Mean parental ages of children in the study series were compared with those of all children born in Germany during 1990/1991-2019.  This is not an appropriate comparison.  We are not told the upper age limit for inclusion in the trials, but it is unlikely to have been lower than 15 years.  In that case, children in the study series could have been born at any time from 1975 onwards, but the control data do not incorporate information on parental ages for births during 1975-1989.  Also, many children born during 2006-2019 who would eventually develop Wilms tumour would not have done so by the end of 2020.  The correct analysis would take into account the distribution of birth years, and preferably region of birth (or failing that, region of residence at diagnosis) among the study series.  This is important because it is highly unlikely that mean parental ages of children born in Germany have remained the same throughout 1975-2019, or even 1990-2019.  There may also be differences in parental ages between regions of Germany.  Why do the very basic comparisons between 1990-1997 and 2010-2018 appear at the end of the Discussion rather than in the Results?  They certainly do not adequately control for fluctuations in parental age patterns across the study period.  The fact that incidence of Wilms tumour did not change significantly over the study period is far from conclusive evidence that there was no parental age effect, since prevalence or magnitude of many other possible risk factors could also have changed over the same period.

3.  There is no information on the proportion of all children with renal tumours overall or with Wilms tumour in particular who were enrolled in the relevant clinical trials, still less on whether the proportion varied by year of diagnosis, age at diagnosis, or region of residence at diagnosis.  This information should be available from the population-based German Childhood Cancer Registry and would enable the authors to assess how representative their study series is of all children with Wilms tumour diagnosed during the study period.

4.  Does the fact that the case series was derived from clinical trial databases mean that all diagnoses were based on central pathology review?  If so, this should be mentioned as a strength of the study.

5.  Case numbers were below 100 for every non-Wilms tumour type.  Discussion should mention that only large differences in parental ages compared with the general population could likely be detected.  Alternatively, the study might be restricted to Wilms tumour only.

6.  Was the study series restricted to children who were born in Germany?  If country of birth is known, then that restriction should be applied.  If it is not known, that should be mentioned as a weakness of the study. 

7.  Results, end of paragraph 2.  The comparison was apparently for all renal tumours, so this sentence should refer to childhood renal tumours overall rather than specifically to Wilms tumour.

8.  Why did analysis jump straight from all renal tumours combined to histological subtypes of Wilms tumour, without any analysis of Wilms tumour (all subtypes combined)?  At the very least, such an analysis would facilitate direct comparison of results with previous studies.

9.  Paternal and maternal ages are quite highly correlated in most populations.  The analyses do not appear to take this into account, and this should be mentioned in the Discussion.

10.  Investigation of outcome in relation to parental age by considering only deceased patients must exclude all but the earliest deaths among the most recent cohorts.  It is stated that the results were confirmed by Cox regression but no details of the model are given.  If survival has changed over time, then outcome, parental ages, birth year and diagnosis year will be interrelated.  The correct analysis would be multivariable, including diagnosis year and birth year as variables.  

11.  Introduction, paragraph 1, final sentence.  To leave open the possibility that no external factors promote development of childhood renal tumours, I suggest rewording as “It is unclear which external factors, if any, promote …”

12.  References 2, 9 and 11.  Details need to be corrected, especially regarding names of authors.

Meaning is generally clear, but quick review of the final version by a native English speaker might help to make the text more idiomatic.  In paragraph 1, for example, sentence 1 would read better as “… has been successfully treated …” and sentence 3 as “The molecular genetic background is not known for all patients.”

Reviewer 2 Report

1The study compared parental ages of childhood WT patients with general German population and assessed the role of parental age on survival of WT patients.

1

1.       More information from results showed be presented in Abstract.

2.       The authors need to give brief summary of previous findings in the Introduction.

3.       Brief information on cases should be presented in the Method, for example age distribution.

4.       It would be helpful if authors could provide a flow chart showing cases and controls selection procedure.

5.       Any differences between included and excluded cases? Were cases excluded mainly because of nationality?

6.       How were controls selected? i.e., selection criteria or matching criteria. How to ensure that the National dataset does not include WT patients?

7.       May offspring of controls have non-WT cancers? Is it possible for authors to include non-cancer controls only? Similar question for the symptom-related analyses.

8.       What is the rationale on the association between parental age and patient survival? What variables were controlled in the Cox model?

9.       In the study, fathers of overall WT cases and mothers of mixed WT cases were younger than those of general population. Could authors obtain other characteristics to further explore?

Minor:

1.       Delete references in Abstract and number references from #1 in the text.

2.       Number tables in the sequence of their appearance

3.       Please provide exact p-values instead of >0.05

4.       Please be consistent in format, such as decimal points and thousand separators.

Reviewer 3 Report

This is a large-scale observational study of the association between Parental Age and Childhood Renal Tumors. ALthough the novelty was limited, the negative result may provide an evidence of this question. Overall quality was moderate.

1 Was Wilms tumor the only tumor in the results? There was not a tumor inclusion criteria in the method.

2 There was not a Cox regression result.

3 How did authors handle patient data (age, gender, syndromes, tumor histology, local and overall stage, surgery, chemotherapy, radiotherapy, complications, and outcome)?

Reviewer 4 Report

This manuscript describes an observational study with the goal of clarifying the impact of parental age on the occurrence of childhood nephroblastoma. I believe some clarity in the description of the study design and participants will improve the presentation of your work.

I recommend following the strobe statement when describing your study:

von Elm E, Altman DG, Egger M, Pocock SJ, Gøtzsche PC, Vandenbroucke JP; STROBE Initiative. The Strengthening the Reporting of Observational Studies in Epidemiology (STROBE)statement: guidelines for reporting observational studies. J Clin Epidemiol. 2008 Apr;61(4):344-9. PMID: 18313558

This includes adding commonly used terms into your article title to define the study design.

Following along this theme of clarity, it is recommended to clearly state the study design at the outset of the materials and methods section. I believe what you are describing is a case-control study with unmatched controls where the cases came from several prospective, multicenter trials and controls come from a dataset derived from the entire population of live births in Germany describing the population. 

To clarify, your study design methods I would recommend clearly stating the study design first, then describing the available data sources with inclusion/exclusion criteria and how your cases and controls were identified. You can then be consistent throughout the manuscript referring to cases and controls. You can then drop the term “test set” which I find confusing in this manuscript. It implies to me that you are doing an intervention rather than describing an observational study.

Are the case participants included within the live births of Germany data or were you able to identify and remove them from the controls. This should be specified and discussed if it is not possible to determine this. Were all the kidney tumor patients live births? If the case patients are found within the Germany live births data, then, technically these two data sets are not independent. In this case, it may be more appropriate to do the one-sample t-test testing if the values in the case data set differ from the population (control) dataset, which would be the Germany live birth data set. It does look to me from your message that however, the two data sets years don’t overlap completely. But I would like discussion of whether the case data set and the live births in Germany dataset are indeed independent from each other.

In table one you have a column labeled cases. Are the counts in the control data set referring to cases of kidney tumors? I did not understand that this was so and us I would just label that column ‘n’.

The title for Table 2 is confusing to me is the age comparison from cases to controls, stratified by parents' gender? The title should clarify this as follows for Table 3 as well. 

In table four there is no reason to abbreviate test and control in the table. This is confusing, and in fact, when I first read the table, I was wondering why gene alleles were coming into play. 

Additional comment

References 1 and 2 should be included in the first sentence in the second paragraph of the introduction lines 41-42, but removed from the abstract.

There are a few awkward sentences, such as the one in lines 38-39. I am not sure if this is due to English language familiarity or the effort to appropriately paraphrase previous work.